# Experimental Study on Mechanical Properties of High Temperature Granite with Different Cooling Methods

Jin-Song Zhang [ORCID], Yu Lu *, Jian-Yong Pang and Yi-Shun Bu

School of Civil Engineering and Architecture, Anhui University of Science and Technology, Huainan 232001, China; 2015004@aust.edu.cn (J.-S.Z.); jypang@aust.edu.cn (J.-Y.P.); 2021076@aust.edu.cn (Y.-S.B.)
* Correspondence: a1299189627@gmail.com; Tel.: +86-152-5204-8355

**Featured Application: Studying the mechanical properties of high-temperature rocks after natural cooling can give us a better understanding of the mechanism of high-temperature effects on rocks. However, in some practical projects (such as geothermal energy mining), the high temperature rock body will be rapidly cooled by water, and its properties will change. Therefore, it is necessary to study the mechanical properties of rocks with different cooling methods, so as to provide theoretical reference for practical engineering.**

**Abstract:** Due to various factors, high-temperature rocks are often affected by different cooling methods. They will lead to changes in rock mechanical properties, which firmly connect with the safety and stability of practical projects. At present, there are many studies of the microscopic characteristics of rocks under different cooling methods, but few on the mechanical characteristics of rocks under the same conditions. Therefore, it is necessary to carry out experimental research on the mechanical properties of rock under different cooling methods. The study shows that the deterioration degree of the sample increases gradually with temperature rising and the cooling methods have different effects on the change of the sample in mass, volume and density. The stress–strain curve of the sample is divided into the crack compaction stage, elastic deformation stage, nonlinear deformation stage and failure stage. The peak strength of the naturally cooled sample is higher than that of the water cooled. The peak strength comes with a trend that goes up first, and then down with the temperature increasing. The uniaxial compression failure of the sample under uniaxial action is tensile failure. The failure characteristic of the sample is influenced both by the cooling modes and the temperature.

**Keywords:** rock mechanics; high temperature; granite; different cooling methods; mechanical properties



## 1. Introduction

In nuclear waste disposal, geothermal energy development and so on, rock mechanics' problems often occur after high temperature. The related parameters of rock after high temperature are the main basis for engineering stability analysis and design. At present, many scholars have carried out relevant research on the mechanical behavior of rock after high temperature. High temperature causes thermal expansion, thermal reaction and thermal stress of minerals inside the rock, which all lead to cracks inside the mineral and changes in the rock microstructure, further to changes in the rock-related properties [1,2]. With the increase in temperature, the mass of the rock decreases gradually and this is mainly related to the change of water occurrence inside the rock. During the heating process, the bound water, crystal water and mineral water inside the rock gradually precipitate with the increase of temperature [3,4]. Zhu et al. [5] found that in the case of temperature rise, the volume and specific heat capacity of granite, which had withstood high temperatures,

increased. The longitudinal wave velocity, mass, density, thermal conductivity and thermal diffusion coefficient of it decreased. Through thin section observation, it was found that there were no obvious thermal cracks in the rock samples after different temperature treatments, but many thermal cracks appeared between the particle cements [6]. At 500~600 °C, the linear expansion coefficient of quartz increased rapidly in both vertical and parallel to the c axis direction, which was mainly due to the α-β phase transition of quartz at about 573 °C, and that the volume of quartz increased by 2.4% [7,8]. The related mechanical properties of granite treated by different high temperatures were studied and it was found that 400 °C was a threshold value for performance change [9]. Through the uniaxial compression test of granite after high temperature treatment, Yang et al. [10] had studied the influence of temperature on the crack damage, strength and deformation failure behavior of granite. Zhao et al. [11] had found that the physical and mechanical properties of granite deteriorated significantly after heat treatment. Through the experimental study of granite under different high temperatures, Xu et al. [12] found that the strength and deformation characteristics of granite samples gradually became stronger, with the increase of confining pressure.

A series of studies were carried out on the physical and mechanical properties of high temperature rock. Most of the scholars have studied the natural cooling of high temperature rock, but that often involves the rapid cooling of water in practical projects, such as geothermal energy mining. Therefore, some scholars have carried out relevant studies to conduct natural cooling and water cooling treatments on granite. When the heat treatment temperature is 25–200 °C, the cooling method has little effect on the physical properties and the wave velocity of different granite samples. With the increasing temperature, the parameter attenuation gradually intensifies, and the water cooling method has a greater effect [13]. Cui et al. [14] found that, with the increase of temperature, the wave velocity, elastic modulus and linear of granite after natural cooling decreased. Zhang et al. [15] studied the meso-damage of granite under different cooling methods. Combined with SEM study, it was found that, with the increase of temperature, the damage to the granite was more serious. The thermal damage of granite is the combined influence of the mechanism of mineral particle expansion and shrinkage, cold shock and thermal physical and chemical changes. Wang et al. [16] studied the tensile strength of granite by different cooling methods. They found that the tensile strength of granite decreases and the rock fracture surface changes to rough and tortuous with the increase in temperature. The cooling effect is more obvious when it comes to water. Although the related properties of rock with different cooling methods were studied, there still remain fewer studies on the uniaxial mechanical properties of rock with different cooling methods. That is what needs to be studied further.

According to all of those mentioned above, relevant engineering rock mass is not only affected by high temperatures, but also by water cooling in the process of reservoir fracturing and mining. Many experts and scholars have conducted relevant research on the changes in rocks' micro properties under different cooling methods, but there is little research about the effects on the changes of rock mechanical properties after different cooling methods. The change of rock mechanical properties directly affects the safety of underground engineering. Therefore, it is necessary to study the change of mechanical properties of samples after different cooling methods. In this paper, the granite after high temperature treatment is taken as the research object. We first treat the samples with different cooling methods and then conduct uniaxial compression test on them. On the basis of discussing the influence of temperature on rock mechanical properties, the evolutionary law of the mechanical properties of different cooling methods is further analyzed.

## 2. Materials and Methods

### 2.1. Sample Processing

The granite samples for this test were taken from Macheng, Hubei Province, as shown in Figure 1. The surfaces of the samples were dense and without cracks. To prevent the

problem of sample dispersion, the samples were taken from the same granite. The samples were made into a standard cylinder of 100 mm × 50 mm according to the international Rock Mechanics Standard, which have an aspect ratio of 2:1.

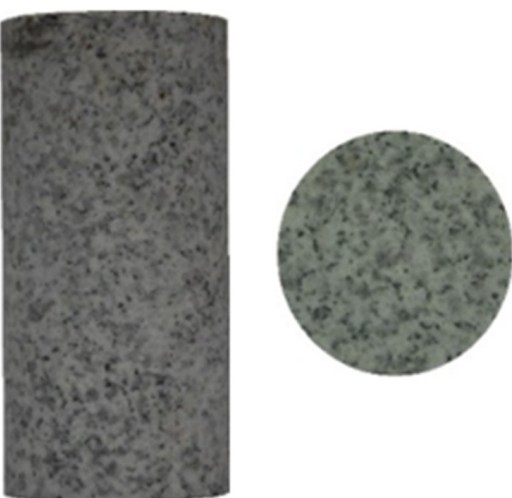

**Figure 1.** Granite Sample.

### 2.2. Test Schemes

The temperature of dry-hot rock mining is 150–650 °C [17]. The temperatures set in the experiment were 200, 300, 400, 500 and 600 °C. The cooling methods were natural cooling at room temperature and water cooling. The specific schemes were as follows:

1. The mass and size of the samples were measured. Then the granite samples in the natural state were treated with high temperature by being heated in a high temperature furnace. In order to avoid cracks caused by a too rapid heating rate, the heating rate was set to 5 °C/min. After heating to the target temperature, the sample was kept warm for two hours to ensure uniform heating. This paper picked three granite samples for each group to undergo the heating treatment and five groups of tests in total were carried out;

2. After the heat preservation, half of the samples were taken out at room temperature for natural cooling. The other half were taken out for cooling in normal temperature water until they reached room temperature (sample immersed). The size and mass of samples were measured, which were treated by different cooling methods, and were then stored in a sealed bag waiting for testing;

3. Samples treated by different cooling methods were used in uniaxial compression tests and they were loaded in the displacement control mode (0.06 mm/min rate) until the sample was damaged. The compressive strength and stress–strain curves of the samples were recorded during this process.

### 2.3. Test Equipment

The experimental samples were made in the SGM resistance high temperature furnace. The heating rate can be controlled by adjusting the target temperature and heating time. The heating range is 0–1200 °C and the display accuracy is ±1 °C/±0.1 °C. All of the mechanical properties tests were carried out on the WAW-1000 electro-hydraulic servo universal testing machine in the State Key Laboratory of Mine Mining and Disaster Prevention and Control of the Anhui University of Technology. The maximum axial load of the device is 1000 KN and the speed range is 0.1–100 mm/min. The accuracy level is 0.5, the piston stroke is 80 mm and the test force resolution is 5 × 105 yards. The device diagram is shown in Figure 2.

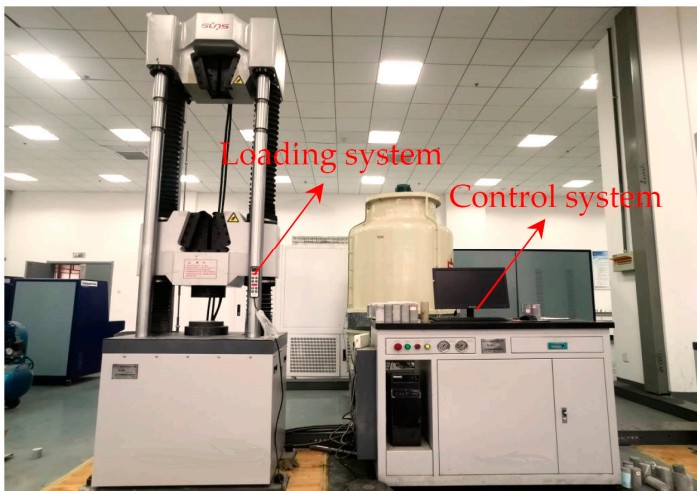

**Figure 2.** WAW-1000 universal testing machine.

### 3. Test Results and Analysis

*3.1. Changes in Physical Properties*

3.1.1. Surface Morphologic Change

Figure 3 shows the surface morphology after natural cooling and water cooling. From Figure 3, we found that under the effect of two cooling methods, the morphology of the sample changes little at 200 °C, 300 °C and 400 °C. The sample particles are closely connected and the morphology is normal. With the increase in heating temperature, fine cracks appear on the surface of the 500 °C and 600 °C samples. This is mainly due to the fact that, with the increase in temperature, the mineral particles in the sample deteriorated to varying degrees, as macrocracks were showing. Furthermore, it can be seen from Figure 3 that more micro cracks appear on the surface of the granite sample, which was cooled by water under the high temperature conditions and the surface of the rock is even rougher. It is considered that the granite, after the high temperature treatment, encounters water when the surface cools rapidly, while the temperature inside the sample does not decrease. So, there is a temperature difference producing both inside and outside stresses, which leads to the temperature stress action and it further causes more microcracks on the surface of the samples.

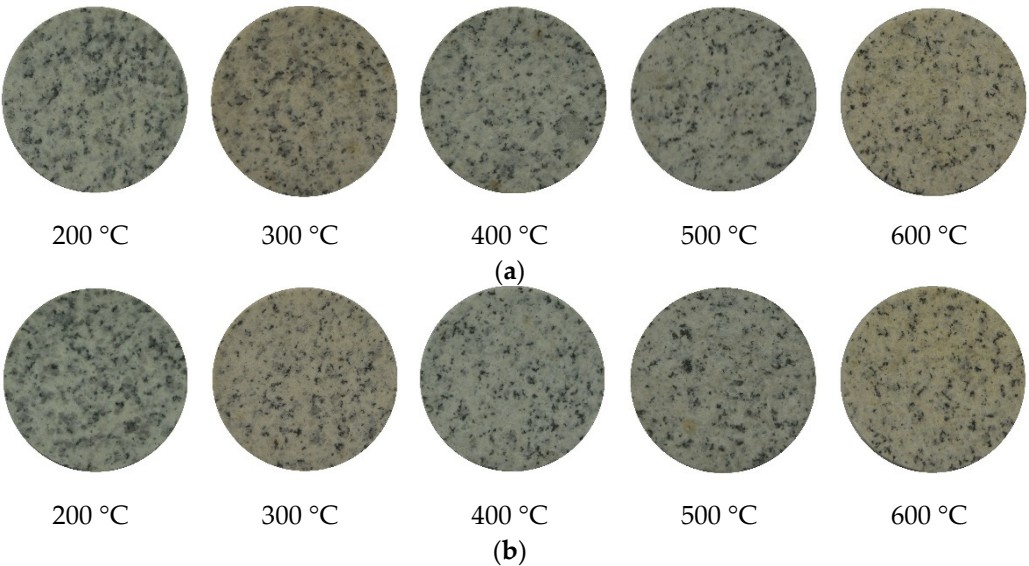

**Figure 3.** Surface morphology of granite with different cooling methods. (**a**) is the surface morphology of granite under natural cooling; (**b**) is the surface morphology of granite under cooling in water.

### 3.1.2. Mass Changes

Temperature and cooling methods often lead to changes in the internal particles of the samples, affecting the mass of it. In order to study effectively the influence of different cooling methods on the mass of high-temperature granite samples, the mass change rate m is introduced, to study the mass change of the samples [18]. As shown in Equation (1):

$$\Delta m = \frac{m_0 - m_1}{m_0} \times 100\% \tag{1}$$

where $m_1$ is the mass of sandstone samples under different high temperature, g; $m_0$ is the mass of sandstone samples before different high temperatures, g.

The changes of the sample mass after different cooling methods are shown in Figure 4. From Figure 4, we can see that the change in the sample mass under different temperatures is slight, which is mainly related to the dense particles in the granite sample and the small initial moisture content [4]. With the increase in temperature, the mass change rate of granite samples under natural cooling gradually goes up and the mass of samples gradually goes down. This change is connected to the state of water occurrence in different forms inside the rock. As the temperature goes up, the attached water and structural water inside the samples precipitate and dissipate to different degrees. The mass change rate of granite samples cooled by water gives a trend that firstly goes down and then goes up. When the temperature is 200, 300 and 400 °C, the mass of the sample cooled by the water gradually increases. It is caused by the evaporation of the water inside the samples, which have endured high temperature and the expansion of the space occupied by water under the action of thermal stress. When the samples are cooled by water, the water will quickly fill these spaces, bringing the increase in mass. When the temperature is 500 °C and 600 °C, the mass of the sample decreases stage by stage after soaking in water, and more microcracks are produced under the action of high temperature. When it is cooled by water, there is a certain degree of spalling with the samples, which is a bigger effect than when the water increases the mass.

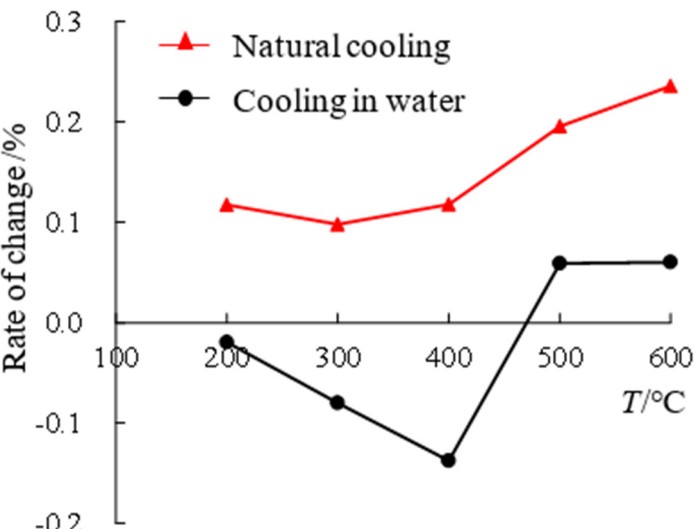

**Figure 4.** Mass change after different cooling methods.

### 3.1.3. Volume Change

Different cooling methods have an influence on the change in the sample volume. In order to better study the volume change of the sample under different cooling methods, the volume change rate $\Delta v$ is introduced, as shown in Equation (2):

$$\Delta v = \frac{v_0 - v_1}{v_0} \tag{2}$$

where: $v_1$ is the volume of samples after different high temperatures, cm$^3$; $v_0$ is the volume of samples before different high temperatures, cm$^3$.

It can be seen from Figure 5 that, under the action of natural cooling, the volume of the granite sample gradually increases with the temperature increasing. This is due to the plastic deformation caused by the expansion of particles in the sample under the action of high temperature. When the sample is cooled, the plastic deformation cannot be restored and that is what causes the increase in the volume. The volume increases more slowly at 400 °C than when the temperature is higher than 400 °C. This is underlined by the rule that the higher the temperature, the greater the plastic deformation of the sample. Under the action of water cooling, the volume of the granite sample decreases first and then increases with the rising temperature. At the high temperature, the internal moisture of the sample will escape continuously and creates a certain deformation space within it. When the sample is cooled rapidly by water, the sample particles shrink, which will occupy the space formed previously. As a result, the sample volume is reduced. When the temperature is higher, the sample volume increases under the action of water cooling. The reason behind this is that the higher the temperature, the greater the plastic deformation, and the deformations cannot dwindle, which makes the volume increase.

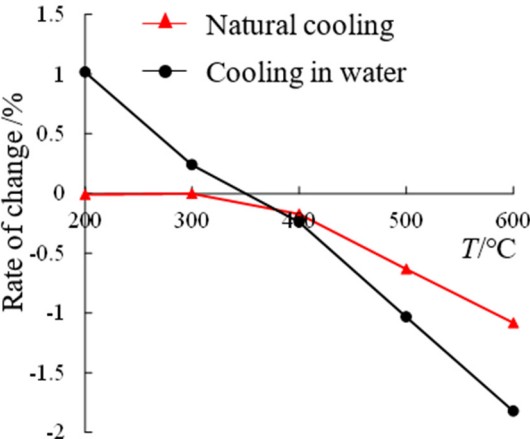

**Figure 5.** Volume change under different cooling methods.

3.1.4. Density Change

In order to explore further and study more deeply the influence of different cooling methods on the physical properties of the sample, the density change of the sample under different cooling methods is studied. The density change rate $\Delta\rho$ is introduced:

$$\Delta\rho = \frac{\rho_0 - \rho_1}{\rho_0} \times 100\% \tag{3}$$

where: $\rho_1$ is the density of samples after different high temperatures, g/m$^3$; $\rho_0$ is the density of samples before different high temperatures, g/m$^3$.

The change of the sample density under different cooling methods is shown in Figure 6. The change rate of the granite sample density under natural cooling gradually increases with the increase in temperature. This is because the change degree of the sample mass is greater than that of the volume under natural cooling. When the treatment temperature is lower than 400 °C, the density of the sample under water cooling is higher than that before the high-temperature treatment, which is caused by the increase in sample mass and the reduction in volume due to cold shrinkage. When the treatment temperature is higher than 400 °C, the sample density decreases gradually with the temperature rising. This is because the granite sample is relatively dense and the mass changes lightly under the action of high temperature. The particles expand greatly when they undergo the temperature operation and under the combined effects of all of the above, the sample density increases gradually.

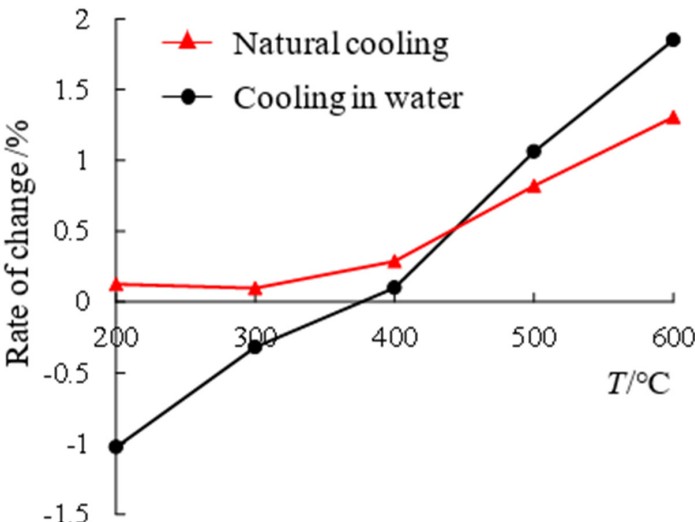

**Figure 6.** Density change under different cooling methods.

*3.2. Changes in Mechanical Properties*

3.2.1. Stress–Strain Curve

In order to analyze the stress–strain characteristics of granite under different cooling methods, the stress–strain curves related are obtained, as shown in Figure 7. The stress–strain curve is divided into the following four stages: (1) In the crack compaction stage, there are irregular cracks inside the natural rock and the cracks in the samples are compacted at the initial loading stage. The stress–strain curve of the samples at this stage shows a nonlinear change; (2) In the elastic deformation stage, the deformation of the samples can be restored with the load occurring. The stress–strain curve shows a linear change; (3) In the nonlinear deformation stage, the internal damage of the samples begins to increase gradually following the loading. The strength of the sample increases gradually until the peak, which is the maximum limit that the samples can bear; (4) In the failure stage, when the sample reaches the peak bearing capacity, damage starts to show. The internal cracks of the samples happen rapidly and penetrate. Macrocracks begin to appear on the surface of samples and the samples are on the verge of being damaged.

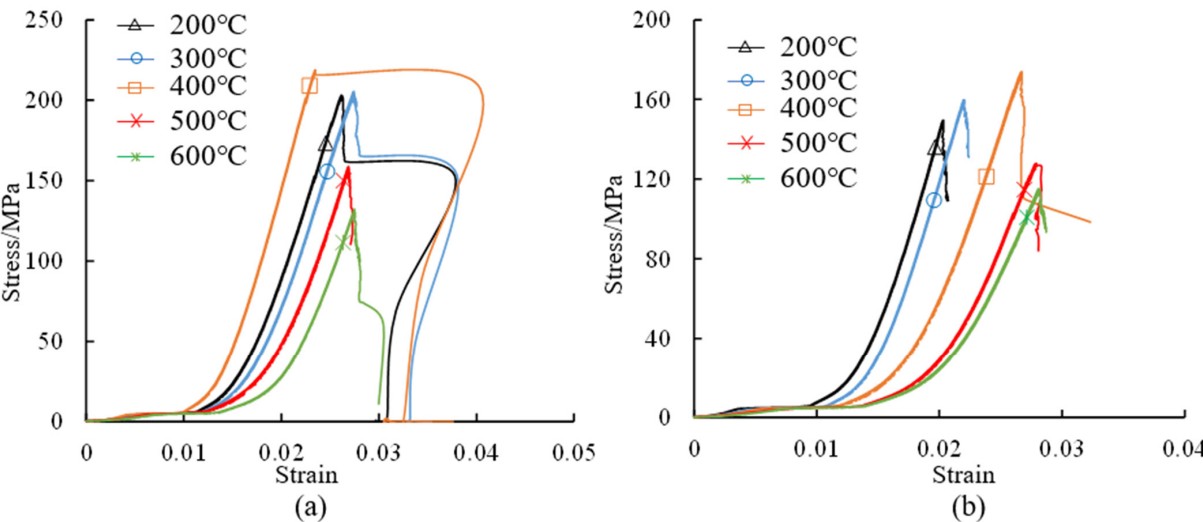

**Figure 7.** Stress–strain curves of samples under different cooling methods. (**a**) The stress–strain curve of the sample in natural cooling mode; (**b**) The stress–strain curve of the sample cooled in water.

### 3.2.2. Strength Characteristics

Figure 8 shows, under different cooling methods, the variation curves of uniaxial compressive strength of granite with temperature change. It can be seen from the figure that, through natural cooling and water cooling, the peak strength of the samples can form into two stages. The temperature threshold is 400 °C. The first stage is from 200 °C to 400 °C and the strength of the samples increases with the temperature at a high level under different treatment methods. At this stage, the water in different forms existing in granite samples slowly escapes after the high temperature. The expansion of particles inside the sample makes these cracks and pores close again. The deformation and crack propagation caused by thermal stress are suppressed to a certain extent. The second stage is from 400 °C to 600 °C. In this stage, the peak strength of the sample decreases step by step with the increase in temperature. As the temperature rises, the mineral particles in the sample keep expanding uncoordinatedly. This leads to a greater thermal stress and more microcracks inside the sample. The temperature is higher, and the phenomenon is more obvious, which accelerates the penetration of the cracks in the sample, but the strength is opposite.

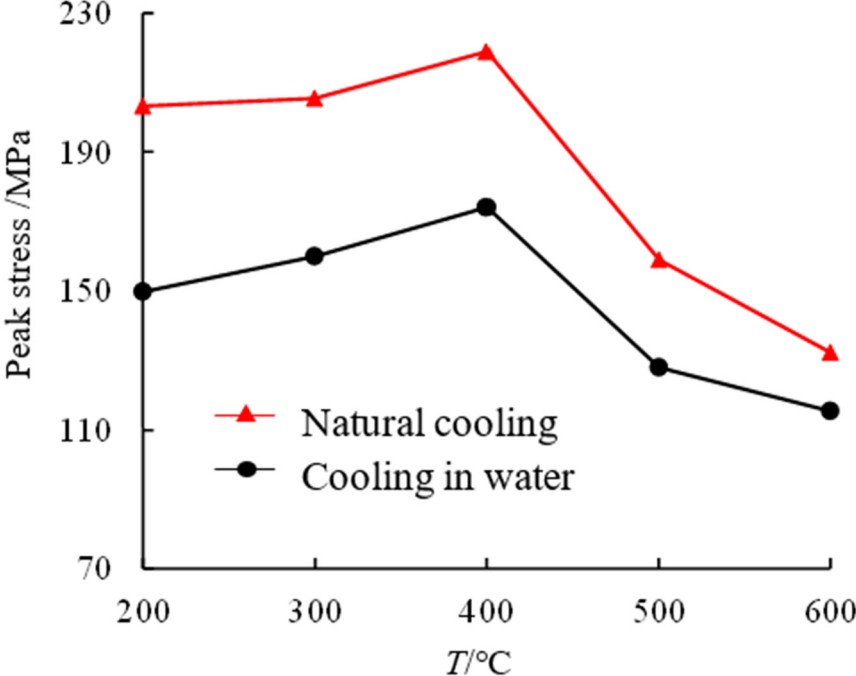

**Figure 8.** Change of peak strength of sample after different cooling methods.

By comparing the compressive strength changes of the samples under different cooling methods, it can be seen that the compressive strength of natural cooling samples at different temperatures is higher than that of water cooling. We can conclude that the degree of deterioration of rock strength by the water cooling method is greater than that of natural cooling. The influence of cold shock damage is greater. High temperatures will change the mineral particles inside the rock to some extent. The internal particles will recrystallize after the cold shock, which will cause more internal pores in the rock [19]. Further, with the samples cooled by water, the periphery of the samples is cooled rapidly, but the interior remains at a high temperature, which causes the temperature difference between the inside and outside of the samples to produce tensile stress. The more serious the damage is, the lower the strength of the sample will be.

The elastic modulus and deformation modulus of the sample can nicely reflect the deformation characteristics. In this paper, the elastic modulus parameters of rock are taken from the partial slope of the approximate straight line in the axial direction—the stress–strain curve of the sample, whose slope of the straight line is 40–60% peak stress. Figure 9 shows the rule of elastic modulus changing with the temperature under different

cooling methods. It can be found from Figure 9 that the elastic modulus of the sample after different cooling methods changes slightly at first, and then decreases gradually as the temperature goes up. This suggests that, as the temperature rises, the change of particles inside the sample attracts the reducing resistance to deformation. The difference in the elastic modulus between different cooling methods is small when it is at a low temperature. However, when the temperature become higher, the elastic modulus of the samples under natural cooling is higher than that under the water cooling. Note that, when the temperature is lower, natural cooling and water cooling have a smaller effect on the deformation resistance of the sample. The deformation characteristics of rock can be expressed by deformation modulus. The secant slope of any point on the stress–strain curve connected with the origin coordinate represents the deformation modulus corresponding to that point. Here, the deformation modulus is taken from the stress–strain ratio of the corresponding point of half of the peak stress intensity. Figure 10 shows the curves of the deformation modulus following with the temperature change in different cooling methods. From Figure 10, it can be seen that the deformation modulus of the samples under different treatment methods changes little as the temperature goes up, which is different from the change in the elastic modulus at the same up situation. The reason for this difference is that the deformation modulus needs to take the compaction stage into consideration.

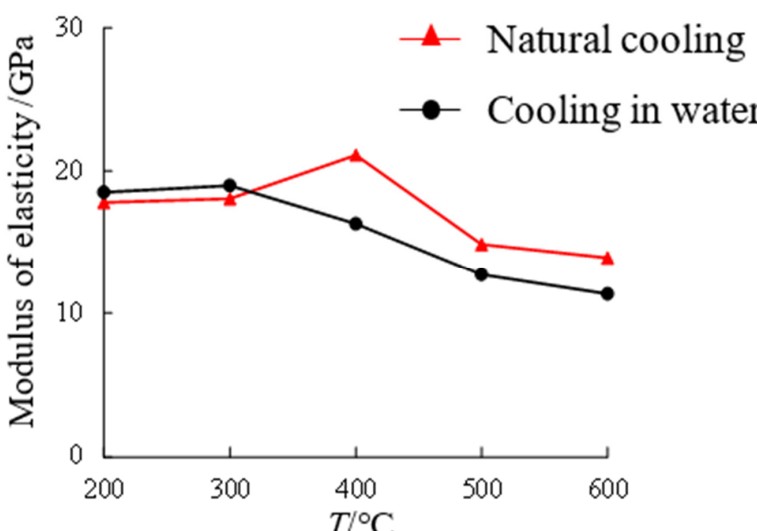

**Figure 9.** Change of elastic modulus of samples after different cooling methods.

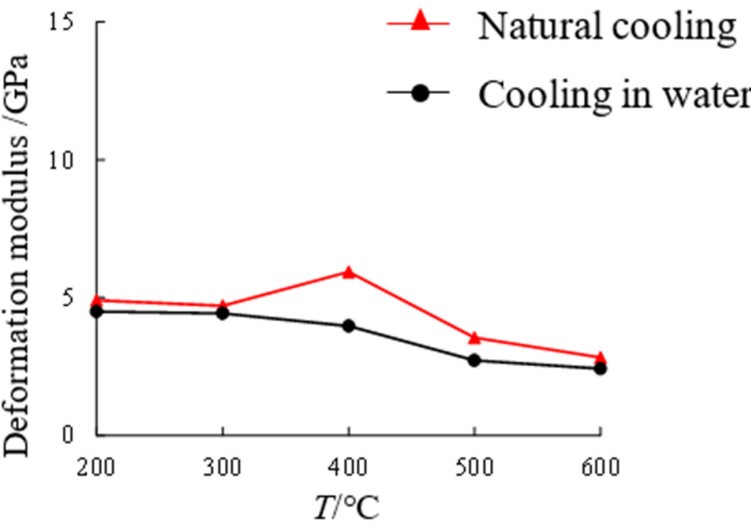

**Figure 10.** Change of deformation modulus of sample after different cooling methods.

### 3.3. Failure Mode Analysis

The failure mode of rock can well reflect the failure degree of rock. The study of the failure mode can better help to understand the deformation characteristics of the sample in the loading process. Figure 11 shows the failure mode of a sandstone sample under uniaxial compression through different cooling methods.

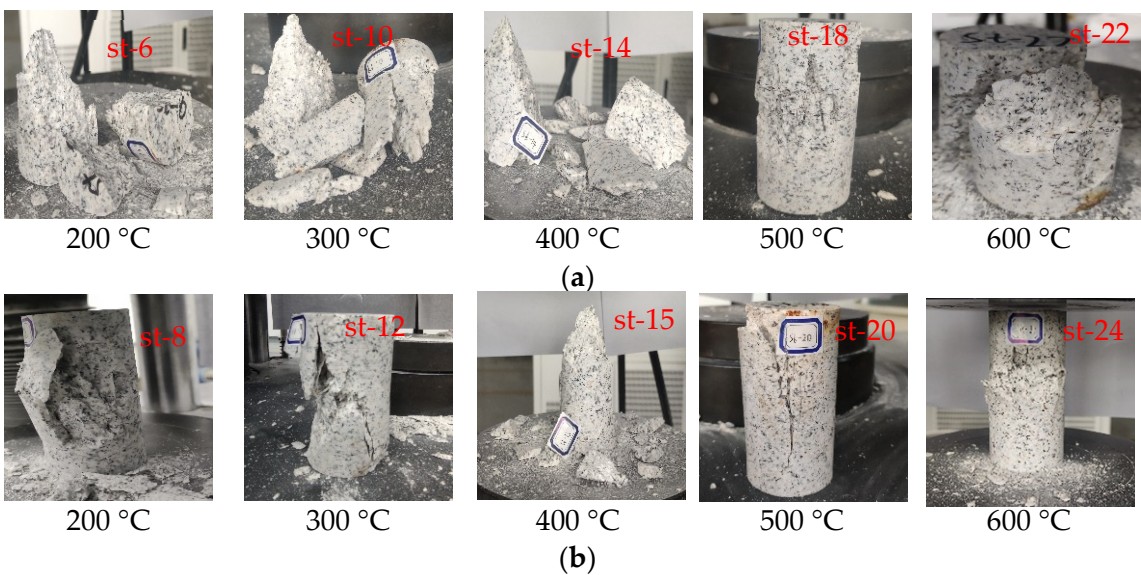

**Figure 11.** Failure modes of granite with different cooling modes. (**a**) Failure modes of granite under natural cooling; (**b**) Failure modes of granite under cooling in water.

Figure 11 demonstrates that the failure mode of the sample under uniaxial action is axial tensile failure, but the cooling mode and temperature affect the failure mode of samples. With the increase in temperature, the failures of samples under natural cooling and water cooling appear to follow the same trend. From Figure 11a,b, it shows that, at 200 °C to 400 °C, the fracture degree of the sample expands gradually with the temperature rising under uniaxial action, leading to more complete massive debris and showing typical brittle failure. At 500 °C to 600 °C, the samples do not completely peel off after failure and there is also a good integrity of the sample. Under the condition of different cooling methods at 500 °C, there are more tensile cracks coming with the sample and a phenomenon that granular fragments falling from the edge of it. When the temperature comes to 600 °C, no large crack shows on the sample. With the influence of the high temperature, the final fracture mode of the sample is affected by thermal damage cracks. The distribution of the microcracks is dense and the particles on the damaged side of the sample are loosely connected. Comparing the failure modes of the samples after different cooling methods, it follows that the failure degree of the natural cooled samples is higher than that of water cooling. The structure of samples after failure is incomplete at 200 to 400 °C. However, the samples are looser inside after damage, with a large number of broken slag being produced and flakey particles presenting.

## 4. Discussion

Different cooling modes of high temperature rock mass are complex processes, and natural cooling and water cooling have different effects on rock mass. Under the action of high temperatures, the rock structure, mineral composition and grain cementation will change [6], affecting the related properties of the rock. During the high temperature treatment of rock, due to the different mineral composition and thermal expansion coefficient of granite, thermal stress is generated between the internal particles of granite, resulting in different types of thermal cracks in the high temperature rock mass. For high-temperature rocks under natural cooling, the high temperature causes plastic deformation of the internal

particles of the rock, and the internal minerals shrink and deform in an uncoordinated manner, filling the gaps between some of the particles. On the other hand, for the high-temperature rock cooled by water, the sample is rapidly cooled, and more thermal cracks are generated in the internal particles. The process is shown in Figure 12.

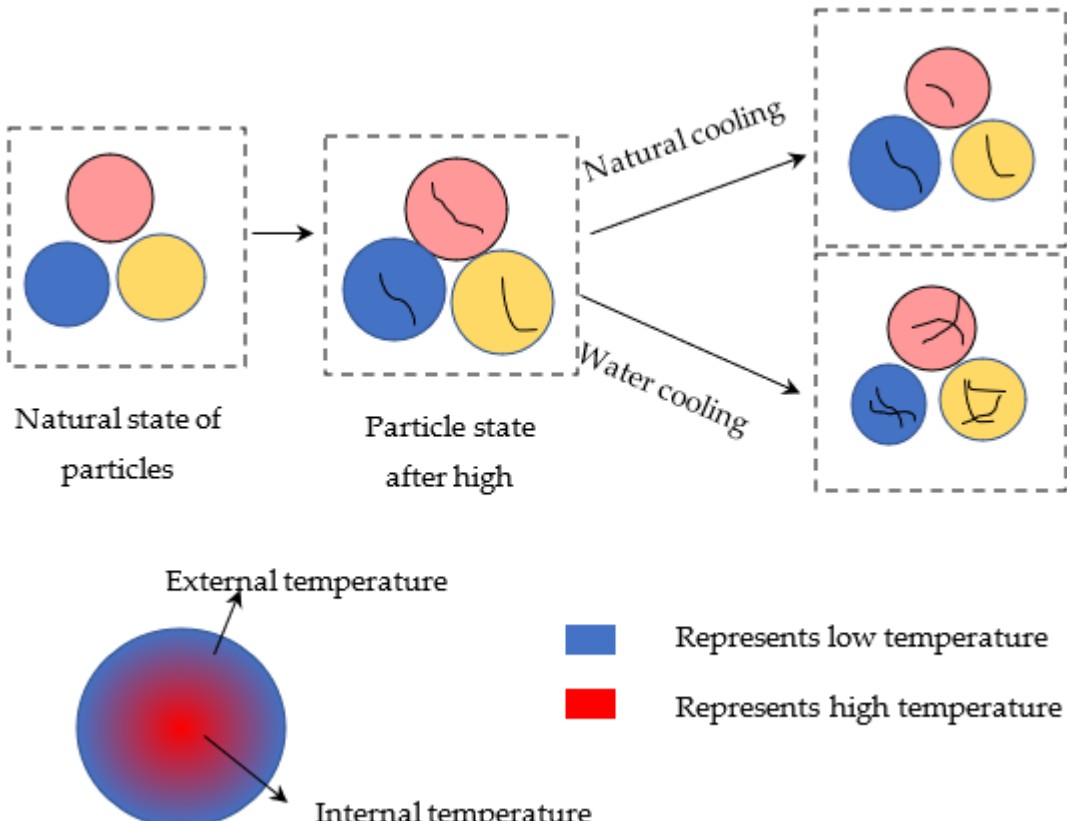

**Figure 12.** Damage process of granite sample under different cooling modes.

Furthermore, the cooling temperature of the sample under the action of natural cooling and water cooling presents a gradient change, and the outer part of the high temperature rock cools first, and then the inner part gradually cools. In this process, due to the change of the temperature gradient, the temperature difference between the inside and outside of the sample is generated, thermal stress is generated between the particles, and with the increase in thermal stress, more thermal cracks appear inside the sample. Because the high-temperature rock cools faster under the action of water cooling [20], the temperature gradient of the sample after being cooled by water is more obvious, which is the reason for more cracks in the water-cooled sample. The sample-related properties are worse than those of the naturally cooled sample, and the gradient changes are shown in Figure 12.

After different cooling methods, the strength of the sample increases before 400 °C, and decreases after 400 °C. Therefore, reducing the temperature when the actual engineering temperature is above 400 °C is beneficial to the safety of the actual engineering. The change of the mechanical properties of the sample under the action of natural cooling is better than that of the sample under water cooling. It can be seen from the failure mode of the sample that the natural cooling sample is looser. Therefore, it is necessary to select an appropriate method for construction in real-life engineering.

This paper only considers the mechanical properties and fracture characteristics of the sample with a single cooling cycle in water. It does not the effect of multiple cooling cycles in water on the sample, nor does it consider the effect of dynamic loads in actual engineering. We will continue to carry out related research in future work, and the research results of this paper provide certain reference value for future related research.

## 5. Conclusions

In this paper, uniaxial compression tests of granite samples treated by different cooling methods are carried out to discover the effect of cooling methods on the mechanical properties of granite. At the same time, the mechanism of temperature and cooling methods on the mechanical properties of granite were revealed. The following conclusions can be drawn:

1.  Under different cooling methods, the samples show a deterioration trend little by little as the temperature goes up, and the deterioration degree is greater through water cooling. With the increase in temperature, the sample mass gradually goes down under the effect of natural cooling. However, under the effect of water cooling, it firstly goes up and then down, which is related to the water occurrence state and material change inside the rock. The volume and density of naturally cooled samples come with the phenomenon of increasing gradually, while the volume of water-cooled samples decreases first and then increases;

2.  The stress–strain curves of the samples after different cooling methods include four stages: crack compaction stage; elastic deformation stage; nonlinear deformation stage and failure stage;

3.  The peak strength of the samples after different cooling methods increases first and then decreases with the temperature rising. At different temperatures, the peak strength of samples under natural cooling is higher than that of water cooling. The elastic modulus of the sample first changed slightly but then became smaller little by little and the deformation modulus changes were less obvious with the temperature increasing;

4.  As the temperature rises, the failure degree of samples with different cooling methods has a hallmark trend of going up first and then down. The integrity of the sample is maintained under high temperature treatment. The damage degree of the natural cooling sample is higher, and for the water cooling is looser.

**Author Contributions:** J.-S.Z. wrote the manuscript; Y.L. edited the manuscript; J.-Y.P. designed research methods; Y.-S.B. edited and revised the manuscript; All authors have read and agreed to the published version of the manuscript.

**Funding:** This research received no external funding.

**Acknowledgments:** The authors would like to thank Editor-in-Chief, Editor and the anonymous reviewers for their valuable reviews. They also thank Tian Wenling of China University of mining and technology for his guidance and support.

**Conflicts of Interest:** The authors declare no conflict of interest.

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
