# Peer review of "Experimental Study on Mechanical Properties of High Temperature Granite with Different Cooling Methods"

_applsci, doi:10.3390/app12125968_

Round 1

Reviewer 1 Report

The paper “Experimental Study on Mechanical Properties of High Temperature Granite with Different cooling methods” is interesting. It shows the results of research in the important field of rock mechanics. Moderate language check are required.

Some improvement is necessary, and the things that should be revised are as follows.

  1. In the Abstract highlight what is new in your research comparing to previous works.
  2. In the introduction you refer to the static cases but dynamic are also very important. I suggest you to read e.g. ZdzisÅ‚aw KÅ‚eczek, Zygmunt NiedojadÅ‚o, Edward PopioÅ‚ek, Wojciech SkobliÅ„ski, PaweÅ‚ Sopata, Tomasz Stoch, Artur Wójcik, Dagmara ZeljaÅ›; Mining hazards analysis with simultaneous minig copper ores and salt deposits in LGOM (Legnica-GÅ‚ogów Copper Belt) mines with regard to dynamic influences, Archives of Mining Sciences 2016.
  3. In the “2.1 Sample processing and test equipment” please definitely put significant amount of information regarding equipment.
  4. Add information how many samples did you use for each measurement and statistical interpretation.
  5. Please describe clearly how do you measure and calculate each value.
  6. Deformation modulus is used for rock mass not for samples. Please provide explanation in the text.
  7. Figure 5 Stress-strain curves of granite with different – sentence not finished. Add legend for colours.
  8. There is no Discussion section – add this section.
  9. Discussion is Conclusions.

Author Response

Response to Reviewer 1 Comments

  1. In the Abstract highlight what is new in your research comparing to previous works.

Response: Special thanks to the reviewer for the careful reading, we have revised the part in the Abstract

  1. In the introduction you refer to the static cases but dynamic are also very important. I suggest you to read e.g. ZdzisÅ‚aw KÅ‚eczek, Zygmunt NiedojadÅ‚o, Edward PopioÅ‚ek, Wojciech SkobliÅ„ski, PaweÅ‚ Sopata, Tomasz Stoch, Artur Wójcik, Dagmara ZeljaÅ›; Mining hazards analysis with simultaneous minig copper ores and salt deposits in LGOM (Legnica-GÅ‚ogów Copper Belt) mines with regard to dynamic influences, Archives of Mining Sciences 2016.

Response: Thanks to the reviewers’ opinions. We read the relevant literature suggested by experts, but this paper mainly studies the changes of mechanical properties of underground engineering rock mass under different cooling modes. Under laboratory conditions, it is mainly the change of rock related characteristics after different cooling methods, and There is no dynamic processing in this paper. Therefore, this paper does not include relevant literature when introducing relevant cases.

  1. In the “2.1 Sample processing and test equipment” please definitely put significant amount of information regarding equipment.

Response: Thanks to the reviewers’ opinions. we have revised the part in 2.1 and 2.3.

  1. Add information how many samples did you use for each measurement and statistical interpretation.

Response: Thanks to the reviewers’ opinions. we have revised the part in 2.2.

  1. Please describe clearly how do you measure and calculate each value.

Response: Thanks to the reviewers’ opinions. we have revised the part in 3.2.2.

  1. Deformation modulus is used for rock mass not for samples. Please provide explanation in the text.

Response: Special thanks to the reviewer for the careful reading, we have revised the part in 3.2.2 . The deformation characteristics of rock are usually expressed by elastic modulus, deformation modulus. The stress-strain relationship of rock is discussed through the results of uniaxial compression test. The slope of the secant at any point on the stress-strain curve connected to the coordinate origin represents the deformation modulus corresponding to the stress at that point. In this paper, it is feasible to select the deformation modulus corresponding to 50% of the peak stress of granite sample to study the deformation characteristics of rock.

  1. Figure 5 Stress-strain curves of granite with different – sentence not finished. Add legend for colours.

Response: Special thanks to the reviewer for the careful reading. And the relevant parts have been revised Figure 7.

  1. There is no Discussion section – add this section.

Response: Special thanks to the reviewer for the careful reading. And the relevant parts have been revised.

  1. Discussion is Conclusions.

Response: Special thanks to the reviewer for the careful reading. And the relevant parts have been revised.

Reviewer 2 Report

The paper entitled "Experimental Study on Mechanical Properties of High Temperature Granite with Different Cooling Methods" reported an interesting work on the very important topic of degradation processes of granite materials caused by changes in temperature by varying the cooling temperature. The article is well written and summarizes some information in its figures and tables, but, in my opinion, it is not enough for a complete manuscript. In this respect, the manuscript contains several missing points and needs revisions before it can be published. Therefore, please improve or clarify the following points: 

  1. It is not clear in the introduction what the aim of your research is. You must give more details in the last paragraph of the introduction on page 2, lines 77-81.
  2. How many samples contain each group of temperatures investigated?
  3. 3.1.1. Subsubsection in line 137 needs to have a title.
  4. To investigate the thermal shock beyond mass loss, it is also important to investigate the volume expansion rate and density change. Therefore, please introduce the equation and discuss it in subsection 3.1.2.
  5. Replace "formula" with "equation" in line 157
  6. At Figures 4, 5, and 6, the legend is missing. Please insert it.
  7. Figure 5's title is incomplete.
  8. How many cycles of thermal shock were performed? It is important to know when failure is reached. Also, microscopic observations of granite during destruction will support your data and improve the quality of the paper.
  9. The acknowledgement section need to be rewritten with proper content. It is necessary to acknowledge some financing, University or colleague that helps with analysis or discussion and are not co-author of the paper.

Based on these, I advise the authors to rectify the above-mentioned issues, and I hope to re-evaluate the revised manuscript.

Author Response

Response to Reviewer 2 Comments

  1. It is not clear in the introduction what the aim of your research is. You must give more details in the last paragraph of the introduction on page 2, lines 77-81.

Response: Special thanks to the reviewer for the careful reading, we have revised the part in the introduction.

  1. How many samples contain each group of temperatures investigated?

Response: Thanks to the reviewers’ opinions. we have revised the part in 3.2.2. This paper picked three granite samples for each group to go through heating treatment and five groups of tests in total were carried out.

  1. 3.1.1. Subsubsection in line 137 needs to have a title.

Response: Special thanks to the reviewer for the careful reading. And the relevant parts have been revised 3.1.1.

  1. To investigate the thermal shock beyond mass loss, it is also important to investigate the volume expansion rate and density change. Therefore, please introduce the equation and discuss it in subsection 3.1.2.

Response: Thanks to the reviewers’ opinions. And the relevant parts have been revised.

  1. Replace "formula" with "equation" in line 157

Response: Thanks to the reviewers’ opinions. And the relevant parts have been revised.

  1. At Figures 4, 5, and 6, the legend is missing. Please insert it.

Response: Thanks to the reviewers’ opinions. And the relevant parts have been revised.

  1. Figure 5's title is incomplete.

Response: Special thanks to the reviewer for the careful reading. And the relevant parts have been revised.

  1. How many cycles of thermal shock were performed? It is important to know when failure is reached. Also, microscopic observations of granite during destruction will support your data and improve the quality of the paper.

Response: Special thanks to the reviewer for the careful reading. Multiple thermal cycle impact tests can better understand when the sample reaches the fault. However, this paper is mainly to study the changes of mechanical properties of samples after different cooling methods, so only one thermal shock is carried out. For some reasons, the microscopic observation of the sample is not carried out in this paper.

  1. The acknowledgement section need to be rewritten with proper content. It is necessary to acknowledge some financing, University or colleague that helps with analysis or discussion and are not co-author of the paper.

Response: Thanks for the reviewer’s positive suggestion. we have revised the part.

Round 2

Reviewer 1 Report

1. There is still no information on static and dynamic cases in the Introduction. Please explain in the Introduction what case and why you describe.

2. There is still a lack of the Discussion section.

Author Response

  1. There is still no information on static and dynamic cases in the Introduction. Please explain in the Introduction what case and why you describe.

Response: Thanks to the reviewers’ opinions. The manuscript focus of this paper is the change of mechanical properties of high-temperature rock under the action of different cooling methods. The experimental design is based on the fact that the high-temperature rock body in underground engineering faces different cooling methods, and its related properties will inevitably change. Therefore, it is necessary to carry out related research and design related engineering. In the background are nuclear waste storage, geothermal energy mining. The dynamic load disturbance problem raised by the reviewers is not the focus of this paper, but the reviewers have provided me with a good research direction, and I will carry out related research in the future. At the same time, the introduction has been further revised

  1. There is still a lack of the Discussion section.

Response: Special thanks to the reviewer for the careful reading. the discussion section have been added.

Reviewer 2 Report

 The author has made substantial improvements to this article. The manuscript can be accepted for publication in the present form.

Author Response

Special thanks to the reviewers . We further revised the manuscript and added a discussion section